# Status of Glucocorticoid-Induced Osteoporosis Preventive Care in Korea: A Retrospective Cohort Study on the Korean National Health Insurance Service Database

**DOI:** 10.3390/medicina58020324

**Published:** 2022-02-21

**Authors:** Byung-Wook Song, A-Ran Kim, Min-A Kim, Ho-Seob Kim, Seung-Geun Lee

**Affiliations:** 1Division of Rheumatology, Department of Internal Medicine, Pusan National University Hospital, Pusan National University School of Medicine, Busan 49241, Korea; medicaldiver@naver.com (B.-W.S.); solees17@naver.com (A.-R.K.); 2Biomedical Research Institute, Pusan National University Hospital, Busan 49241, Korea; 3Department of Data Science, Hanmi Pharm. Co., Ltd., Seoul 05545, Korea; mina.kim92@hanmi.co.kr (M.-A.K.); hoseob.kim@hanmi.co.kr (H.-S.K.)

**Keywords:** glucocorticoids, osteoporosis, prevention, bone density

## Abstract

*Background and Objectives:* It is crucial to prevent osteoporosis in patients receiving long-term glucocorticoid (GC) treatment. This study aimed to investigate the frequency and associated factors of preventive care for glucocorticoid-induced osteoporosis (GIOP) in Korea. *Materials and Methods:* Using the Korean National Health Insurance Service database, we identified 37,133 individuals aged ≥ 20 years who commenced long-term (≥90 days) oral GC between 2011 and 2012. High-quality GIOP preventive care was defined as either a bone mineral density (BMD) test, calcium and/or vitamin D supplementation, or prescription osteoporosis medications within 6 months of GC initiation. Multivariable logistic regression models were used to calculate odds ratios (ORs) for associated factors for high-quality GIOP preventive care. *Results:* The mean age was 49.8 years, and 18,476 (49.8%) patients were female. The frequency of high-quality GIOP preventive care was only 3.68% (BMD test, 1.46%; osteoporosis medications, 1.65%; calcium/vitamin D, 1.63%). Increasing age (OR = 2.53, *p* < 0.001; 40–49 years, OR = 3.99, *p* < 0.001; 50–59 years, OR = 5.17, *p* < 0.001; 60–69 years, OR = 8.07, *p* < 0.001; ≥70 years, respectively), systemic autoimmune disease (OR = 3.08, *p* < 0.001), rural residence (OR = 1.19, *p* = 0.046), concomitant hyperthyroidism (OR = 1.58, *p* = 0.007), and malignancy (OR = 1.59, *p* < 0.001) were significantly associated with a higher likelihood of receiving high-quality GIOP preventive care. Male sex (OR = 0.26, *p* < 0.001) and GC prescription in primary care clinics and nursing hospitals (OR = 0.66, *p* < 0.001) were associated with a lower rate of high-quality GIOP preventive care. *Conclusions:* Most Korean patients treated with GC did not receive appropriate preventive care for GIOP in real-world practice. More efforts are needed by clinicians to prevent, screen, and treat GIOP.

## 1. Introduction

Approximately 1% of the adult population worldwide are treated with long-term systemic glucocorticoids (GC), which are widely used for the management of a variety of disorders due to their anti-inflammatory and immunosuppressive properties; these disorders include systemic autoimmune diseases, inflammatory bowel diseases, chronic pulmonary diseases, allergic diseases, hematologic malignancy, and following organ transplants [1,2]. However, long-term use of GC can cause a myriad of adverse effects, among which the most serious one is the reduction of bone density and derangement of bone quality, leading to glucocorticoid-induced osteoporosis (GIOP), the most common form of secondary osteoporosis. GIOP develops in approximately 30–50% of patients receiving long-term GC therapy, and the risk of fracture increases by more than 50% in these patients [1], which imposes a significant clinical burden due to increased morbidity and mortality [3,4]. Although GC can affect both trabecular and cortical bones, fragility fractures occur more commonly in the trabecular bone, such as the lumbar vertebrae [5]. Bone loss occurs rapidly within the first 3 to 6 months of GC initiation and persists at a slower rate thereafter; thus, the risk of fracture peaks at 12 months [4,5]. Fractures can occur even with low daily dose exposure of long-term GC and are independent of bone mineral density (BMD) and age [1]. Thus, GC should be administered at the lowest dose for the shortest period of time, and if long-term glucocorticoid treatment is inevitable, special attention is required to prevent fractures.

Numerous academic societies and national-level specialty groups, such as the American College of Rheumatology [6], the Japanese Society for Bone and Mineral Research, the Korean Society for Bone and Mineral Research [7], the Korean College of Rheumatology (KSBMR/KCR) [5], the French Society for Rheumatology and Osteoporosis Research and Information Group [8], the International Osteoporosis Foundation, and the European Calcified Tissue Society [9], have released guidelines for the management of GIOP. Most guidelines recommend that preventive measures for GIOP, including BMD tests, calcium and/or vitamin D supplementation, and osteoporosis medications, should be commenced within 6 months of GC initiation if daily doses of 2.5 mg to 7.5 mg or more of prednisolone-equivalent GC have to be taken continuously for 3 months or more. In particular, the Korean GIOP guideline (KSBMR/KCR guideline) proposed that fracture risk assessment, such as the Fracture Risk Assessment Tool, should be performed for adults who are using or planning to receive GC therapy. However, it has been suggested that there is a considerable gap in the prevention and treatment of GIOP in real-world practice [3,10,11,12,13,14,15,16,17,18,19]. Existing data on the status of GIOP prevention have been mostly focused on Western countries, and studies on Asians, especially Koreans, are lacking. Therefore, the objective of this study was to investigate the frequency and associated factors of preventive care in patients with a risk of GIOP, using a nationwide claims database in the Republic of Korea.

## 2. Materials and Methods

### 2.1. Study Design and Subjects

The design was a population-based retrospective cohort study using data obtained between 2010 and 2013 from the Korean National Health Insurance Service (KNHIS) claims database, which contains all outpatient and inpatient claims, medication prescriptions, medical procedures, and diagnostic codes defined by the Korean Standard Classification of Disease 10th Revision (KCD-10), along with demographic information, such as age, gender, residence, and healthcare providers’ information. The KNHIS covers nearly the entire Korean population as obligatory social insurance. In accordance with the Personal Information Protection Act, the identification of each insurant was encrypted in the KNHIS system and scrambled before the data were sent to investigators.

We analyzed adult patients aged 20 years and older who had commenced long-term oral GC treatment from January 2011 to December 2012. Long-term oral GC treatment was defined as receiving any dose of oral GC for at least 90 consecutive days. Oral GC included any dose of prednisolone, methylprednisolone, triamcinolone, dexamethasone, betamethasone, deflazacort, and hydrocortiosone, which was available during the study period. The date of the first oral GC prescription was set as the index date. The following patients were excluded from the analysis: (1) those aged less than 20 years; (2) those who received oral GC within 1 year prior to the index date; (3) diagnosis and/or treatment of osteoporosis (KCD-10 codes M80, M81, and M82) or fragility fracture within 1 year prior to the index date; (4) those who had been prescribed oral GC for <90 days; and (5) death or fragility fracture within the first 90 days after the index date. Fragility fractures included vertebral fractures, hip fractures, and distal radius fractures. A vertebral fracture was defined as the following KCD-10 diagnostic codes: S22.0 (fracture of the thoracic spine), S22.1 (multiple fractures of the thoracic spine), S32.0 (fracture of the lumbar spine), M48.4 (fatigue fracture of the vertebra), and M48.5 (collapsed vertebrae, not elsewhere classified) accompanied by >3 outpatient visits or ≥1 hospital admission [20,21]. A hip fracture was defined as the following KCD-10 diagnostic codes: S72.0 (fracture of the neck) and S72.1 (pertrochanteric fracture) with related operations, including open reduction of a fracture, closed pinning, closed intramedullary nailing, total hip arthroplasty, or hemiarthroplasty [21,22]. A distal radius fracture was defined as the following KCD-10 codes: S52.5 (fracture of the lower end of radius) and S52.6 (fracture of the lower end of both the ulna and radius) and related operations, including open reduction of the ulnar or radius, closed pinning of the ulnar or radius, external fixation of the forearm bone, closed reduction of the forearm bone, long arm cast application, and short arm cast application [21,23].

A flow chart of the study is shown in Figure 1; data relating to 37,133 long-term GC users were investigated. The Research and Ethical Review Board of Pusan National University Hospital approved the study and waived the need for informed consent because patient identity was anonymous to investigators who analyzed the KNHIS database and because of the retrospective design of our study (IRB no. 2009-011-095).

### 2.2. Outcomes

The primary outcome of our study was whether long-term GC users received high-quality GIOP preventive care after oral GC therapy. High-quality GIOP preventive care was defined as either one of the following: (1) a BMD test, (2) prescription of calcium and/or vitamin D supplements, and/or (3) prescription of osteoporosis medications, if either one occurred within 180 days after the initiation of oral GC (index date) [17,24]. Osteoporosis medications included bisphosphonates and selective estrogen receptor modulators, which were available during the study period [17,24]. However, estrogens and other hormones were not considered as osteoporosis medications [17,24], whereas teriparatide and denosumab were not available in Korea during the study period.

### 2.3. Covariates

The following demographic and clinical variables were extracted from the KNHIS database: age; residence (urban/rural); institution (tertiary hospital/general hospital/primary care clinic/nursing hospital); initial GC prescriber specialty (rheumatologist/non-rheumatologist internist/non-internist); GC-requiring conditions (systemic autoimmune diseases/chronic pulmonary diseases/others); cumulative GC prednisolone-equivalent dose within the first 90 days after the index date; and comorbidities associated with osteoporosis (hyperparathyroidism/hyperthyroidism/hypothyroidism/malignancies). GC-requiring conditions were determined by the main diagnostic codes at the index date. Systemic autoimmune diseases included rheumatoid arthritis (KCD-10 codes M05 and M06), polymyalgia rheumatica (KCD-10 codes M353), systemic lupus erythematosus (KCD-10 codes M32), other rheumatic diseases (KCD-10 codes M30, M31, M33, M34, M35), and inflammatory bowel diseases (KCD-10 codes K50, K51). Chronic pulmonary diseases included chronic obstructive pulmonary disease (KCD-10 codes J44) and asthma (KCD-10 codes J45 and J46). Comorbidities were determined by the diagnostic codes within 1 year prior to the index date (KCD-10 codes E211, E212, E213 for hyperparathyroidism, E05 for hyperthyroidism, E03 for hypothyroidism, and C00–C97 for malignancies).

### 2.4. Statistical Analyses

All statistical analyses were performed using the Statistical Analysis System (version 9.2; SAS Institute, Cary, NC, USA), and *p*-values less than 0.5 were considered statistically significant. Descriptive statistics were represented as mean ± standard deviation (SD) for continuous variables and frequency with percentage for categorical variables, as appropriate. Group comparisons were analyzed using Student’s *t*-test for continuous variables and the Mann–Whitney U test or Fisher’s exact test for categorical variables, as appropriate. Multivariable logistic regression models, including variables with *p*-values less than 0.5 in univariable models without variable selection, were used to identify factors associated with high-quality GIOP preventive care. Odds ratios (ORs) with 95% confidence intervals (CIs) were calculated to measure the strength of statistical significance.

## 3. Results

The baseline demographic and clinical characteristics of the 37,133 patients who commenced long-term GC therapy are summarized in Table 1. The mean age was 49.8 years and approximately half of the study subjects (49.8%) were female. The majority of patients (84.2%) received oral GC therapy in primary care clinics or nursing hospitals. Only a small proportion of subjects (1.1%) were treated with oral GC by rheumatologists, whereas non-internists prescribed oral GC to approximately two-thirds of the subjects (67.6%). The frequencies of systemic autoimmune diseases and chronic pulmonary diseases were 3.4% and 8.5%, respectively. The mean cumulative GC dose within the first 90 days after the index date was 203.8 mg prednisolone-equivalents.

The frequency of high-quality GIOP preventive care within the first 180 days after the initiation of long-term GC treatment was 3.7%; 1.5% of subjects had received BMD tests, 1.7% were prescribed calcium and/or vitamin D supplements, and 1.6% were treated with osteoporosis medications (Figure 2).

A comparison of clinical variables according to high-quality GIOP preventive care is shown in Table 2. Those who received high-quality GIOP preventive care were significantly older, more likely to be female, more frequently residing in rural areas, had a higher frequency of receiving oral GC in tertiary/general hospitals, were exposed to lower cumulative GC doses, and had a higher frequency of hyperthyroidism, hypothyroidism, and malignancy than those who did not receive high-quality GIOP preventive care. In addition, there was a significant difference in the frequency of prescriber specialty and GC-requiring conditions, according to whether the high-quality GIOP preventive care was received or not. Those who were treated with GC by a rheumatologist or those who were prescribed oral GC for systemic autoimmune disease tended to be more likely to receive high-quality GIOP preventive care.

Appendix A summarize the comparison of clinical characteristics with regard to the presence or absence of a BMD test, calcium and/or vitamin D treatment, and the prescription of osteoporosis medications. The results were almost identical when compared according to each item of the high-quality GIOP preventive care. Compared with those who were younger, male, and cared for in primary care clinics/nursing hospitals, those who were older, female, and cared for in tertiary/general hospitals had a higher frequency of receiving BMD tests, calcium and/or vitamin D supplements, and osteoporosis medications. The mean cumulative GC dose in subjects who underwent a BMD test was significantly higher than in those who did not undergo the BMD test (Appendix A), whereas those treated with osteoporosis medications had a significantly lower cumulative GC dose than those not receiving these medications (Appendix A).

The results of the logistic regression models analyzing factors associated with high-quality GIOP preventive care are presented in Table 3. In univariable analyses, increasing age categories, rural residence, systemic autoimmune diseases, chronic pulmonary diseases, concomitant hyperthyroidism, hypothyroidism, and malignancy were associated with a higher likelihood of high-quality GIOP preventive care, whereas male gender, care in primary care clinics/nursing hospitals, prescription by non-rheumatologists and non-internists, and an increased cumulative GC dose were associated with a lower likelihood of high-quality GIOP preventive care. In multivariable logistic regression models adjusted for all significant covariates in univariable analyses, increasing age categories (40~49 years: OR = 2.53, 95% CI = 2.02–3.17, *p* < 0.001; 50~59 years: OR = 3.99, 95% CI = 3.23–4.92, *p* < 0.001; 60~69 years: OR = 5.17, 95% CI = 4.16–6.43, *p* < 0.001; ≥70 years: OR = 8.07, 95% CI = 6.5–10.03, *p* < 0.001), rural residence (OR = 1.19, 95% CI = 1–1.42, *p* = 0.046), systemic autoimmune diseases (OR = 3.08, 95% CI = 2.49–3.8, *p* < 0.001) and concomitant hyperthyroidism (OR = 1.58, 95% CI = 1.13–2.21, *p* = 0.007) and malignancy (OR = 1.59, 95% CI = 1.24–2.03, *p* < 0.001) showed independent associations with receiving high-quality GIOP preventive care. Male sex (OR = 0.26, 95% CI = 0.23–0.3, *p* < 0.001) and GC prescription in primary care clinics/nursing hospitals (OR = 0.66, 95% CI = 0.57–0.75, *p* < 0.001) were inversely related to high-quality GIOP preventive care.

Factors associated with BMD tests, calcium and/or vitamin D supplementation, and osteoporosis medication prescription are shown in Appendix A. Increasing age categories and systemic autoimmune diseases were related to a higher frequency of all items consisting of high-quality GIOP preventive care, whereas male gender and primary care clinics/nursing hospitals were associated with a lower rate of these items.

## 4. Discussion

This nationwide population-based retrospective cohort study described the epidemiology of preventive management for GIOP in Korean adults. Our data found that the frequency of high-quality GIOP preventive care in the Korean population was less than 4%, suggesting that preventive measures for osteoporosis in patients receiving long-term GC therapy are suboptimal, in line with previous reports in other countries. Aging, rural residence, GC prescription for systemic autoimmune diseases, and concomitant hyperthyroidism and malignancy were found to be significantly associated with a higher likelihood of receiving high-quality GIOP preventive care, whereas male gender and medical care in primary care clinics/nursing hospitals were linked with a lower frequency of high-quality GIOP preventive care. In addition, our data found that a large number of Korean adults were prescribed long-term low-dose oral GC at primary medical institutions by non-internists for a wide variety of diseases, except systemic inflammatory autoimmune diseases and chronic pulmonary diseases, raising concerns about the excessive use of GC in real-world practice.

Our data revealed that the majority of long-term GCs users did not receive GIOP preventive care. As shown in Table 4, the frequency of BMD testing, calcium and/or vitamin D supplementation, and osteoporosis medication use varied between 6% to 44%, 18% to 69%, and 4% to 51.8%, according to previous studies. Although various GIOP guidelines, including Korean guidelines, have been published to date, our data and others suggest that there is a significant gap in the prevention of osteoporosis in those taking long-term GC. In addition, the likelihood of receiving preventive care for GIOP was considerably lower in our study than in other studies. This difference may be attributable to the low awareness of GIOP prevention among clinicians in Korea, but also due to differences in the study design, eligibility criteria for study participants, ethnicity, geographical area, and medical insurance system of each country. Similar to the studies by Majumdar et al. [17] and Curtis et al. [13], we analyzed patients who were prescribed oral GC for ≥90 days regardless of the daily GC dose. The majority of previous studies investigated long-term GC users with a daily GC dose of at least 5 mg of prednisolone equivalent. Therefore, compared with these studies, our study is more likely to have included patients with lower cumulative glucocorticoid use within 90 days after the index date, as well as those with less comorbidity, or with less severity of underlying illness, all of whom are less likely to receive GIOP prevention. In addition, patients with a previous history of osteoporosis or fragility fractures were excluded from our analysis, which may also have affected the low frequency of GIOP prevention. This notion is supported by Ettinger et al., who reported that previous osteoporotic fractures were associated with a greater likelihood of osteoporosis medication being prescribed [11]. We suggest that the application of broad inclusion criteria and strict exclusion criteria for patients in our study may be the main reason for the lower rate of GIOP prevention compared with other studies.

Factors associated with GIOP preventive care were identified in our study. Female sex and aging were associated with a higher likelihood of GIOP preventive care in our data, consistent with previous reports [10,11,16,17,18]. This finding indicates that male patients and younger patients are less likely to receive preventive measures for GIOP. Because chronic exposure to GC can increase the risk of osteoporotic fracture in both men and women as well as in patients younger than 40 years [5,6,25], clinicians should pay more attention to the management of GIOP for male patients and younger patients receiving long-term GC treatment. Although GC prescription by rheumatologists was consistently associated with a higher likelihood of GIOP prevention in previous studies [10,11,13,16,18], this association was not observed in our data. Otherwise, this study found that those who were prescribed long-term GC for systemic autoimmune diseases were associated with a higher frequency of preventive care for GIOP, and similar results were reported by Trijau et al. [18] As suggested by Albaum et al. [3], rheumatologists treat patients with systemic autoimmune diseases and osteoporosis, which have a more chronic nature than other diseases as their clinical specialties. This may explain the high awareness of GIOP prevention when rheumatologists prescribe long-term GC for patients with systemic autoimmune diseases.

Prescriptions of GC at primary care clinics/nursing hospitals were associated with a lower rate of GIOP prevention in our study, suggesting that clinicians in these institutions may have lower adherence to GIOP guidelines than those in tertiary/general hospitals. Thus, academic societies related to osteoporosis in Korea need to raise awareness of the importance of GIOP prevention and provide information on GIOP guidelines for primary care physicians. Unexpectedly, we found that medical care in rural medical institutions was related to a higher frequency of high-quality GIOP preventive care than in urban medical institutions. Because urban overcrowding is serious in Korea, clinicians in urban hospitals tend to see more patients in a limited time than those in rural hospitals, which may explain this finding. However, the exact mechanism for this is not completely understood, and further studies are needed to determine the reasons for differences in the frequency of preventive measures against osteoporosis in urban and rural areas.

Our study had several limitations. First, as we did not include intravenous GC, inhaled GC, and episodic GC therapy (for example, management for frequent gout attacks or acute exacerbation of pulmonary disorders), this study may not have included all patients who are at risk for GIOP and fragility fractures. In addition, the dose and duration of GC recorded in the national claims database may differ from those actually taken by patients [14]. For example, if a clinician prescribes prednisolone to a patient for 4 weeks with a 5mg reduction each week, the patient will take prednisolone for 4 weeks, but the prescription may be recorded in the claims database as if four different doses of prednisolone are prescribed for 1 week. For this reason, we assumed that it was difficult to collect patients on high-dose GC induction therapy in the claims database. Second, BMD tests and osteoporosis medications not reimbursed by KHNIS and over-the-counter calcium and/or vitamin D supplementation were not captured in the KNHIS database, which may result in an underestimation of the frequency of high-quality GIOP preventive care. Third, we could not fully obtain the detailed information on underlying diseases and osteoporosis, including the exact diagnosis, disease severity, and the results of the BMD testing in the KHNIS database.

## 5. Conclusions

In conclusion, the present study demonstrates that the majority of Koreans taking long-term GC did not receive adequate preventive care for osteoporosis and fragility fractures despite a variety of therapeutic agents and effective guidelines that are available for the management of GIOP in Korea; this indicates that there may be a significant public health problem. Although the identification of the exact reasons for the low rate of GIOP prevention in real-world practice is beyond the scope of this study, more efforts are needed by clinicians for the screening, prevention, and treatment of GIOP, especially for men and those who receive medical care in primary care clinics/nursing hospitals and in urban medical institutions. Further studies are warranted to investigate the impact of suboptimal preventive care for GIOP on the burden of fragility fractures.

## Figures and Tables

**Figure 1 medicina-58-00324-f001:**
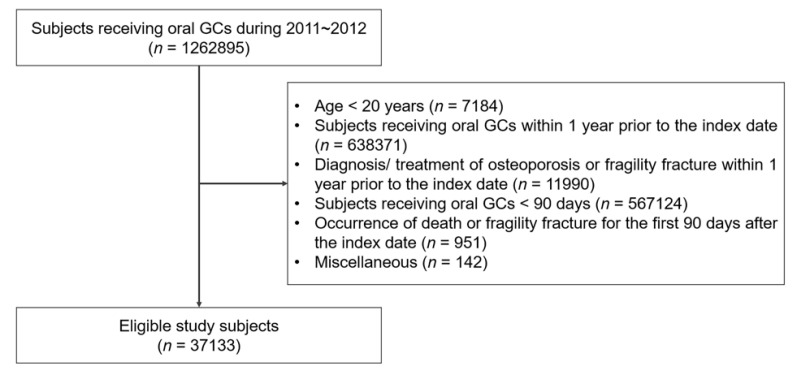
Flow chart of the present study.

**Figure 2 medicina-58-00324-f002:**
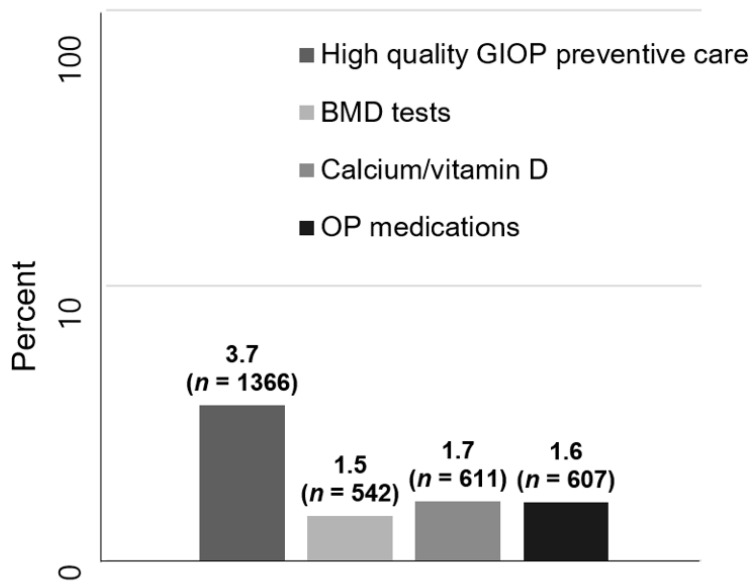
The frequency of high-quality glucocorticoid-induced osteoporosis preventive care.

**Table 1 medicina-58-00324-t001:** Baseline demographic and clinical characteristic of new long-term glucocorticoids users.

	New GC Users (*n* = 37,133)
Age	
<40 years, *n* (%)	10,307 (27.8)
40~49 years, *n* (%)	7818 (21.1)
50~59 years, *n* (%)	8798 (23.7)
60~69 years, *n* (%)	5839 (15.7)
≥70 years, *n* (%)	4371 (11.8)
Age, years, mean ± SD	49.8 ± 15.3
Female, *n* (%)	18,476 (49.8)
Residence	
Urban, *n* (%)	33,899 (91.3)
Rural, *n* (%)	3234 (8.7)
Institution	
Tertiary/general hospital, *n* (%)	5861 (15.8)
Primary care clinic/nursing hospital, *n* (%)	31,272 (84.2)
Initial GC prescriber specialty	
Rheumatologist, *n* (%)	415 (1.1)
Non-rheumatologist internist, *n* (%)	11,608 (31.3)
Non-internist, *n* (%)	25,110 (67.6)
GC-requiring conditions	
Systemic autoimmune diseases, *n* (%)	1274 (3.4)
Chronic pulmonary diseases, *n* (%)	3171 (8.5)
Others, *n* (%)	32,688 (88.1)
Cumulative GC dose *, mg, mean ± SD	203.8 ± 151.4
Comorbidities	
Hyperparathyroidism, *n* (%)	8 (0)
Hyperthyroidism, *n* (%)	579 (1.6)
Hypothyroidism, *n* (%)	929 (2.5)
Malignancy, *n* (%)	1127 (3)

* Cumulative GC dose in the first 90 days after the initiation of GC therapy. GC glucocorticoids, SD standard deviation, C GIOP glucocorticoid-induced osteoporosis.

**Table 2 medicina-58-00324-t002:** Comparisons of clinical characteristics according to high-quality glucocorticoid-induced osteoporosis.

	No High-Quality GIOP Preventive Care (*n* = 35,767)	High-Quality GIOP Preventive Care (*n* = 1366)	*p*
Age			
<40 years, *n* (%)	10,187 (28.5)	120 (8.8)	<0.001
40~49 years, *n* (%)	7588 (21.2)	230 (16.8)	
50~59 years, *n* (%)	8415 (23.5)	383 (28)	
60~69 years, *n* (%)	5545 (15.5)	294 (21.5)	
≥70 years, *n* (%)	4032 (11.3)	339 (24.8)	
Age, years, mean ± SD	49.5 ± 15.3	58.4 ± 13.8	<0.001
Female, *n* (%)	17,443 (48.8)	1033 (75.6)	<0.001
Residence			
Urban, *n* (%)	32,696 (91.4)	1203 (88.1)	<0.001
Rural, *n* (%)	3071 (8.6)	163 (11.9)	
Institution			
Tertiary/general hospital, *n* (%)	5513 (15.4)	348 (25.5)	<0.001
Primary care clinic/nursing hospital, *n* (%)	30,254 (84.6)	1018 (74.5)	
Initial GC prescriber specialty			
Rheumatologist, *n* (%)	374 (1)	41 (3)	<0.001
Non-rheumatologist internist, *n* (%)	11,145 (31.2)	463 (33.9)	
Non-internist, *n* (%)	24,248 (67.8)	862 (63.1)	
GC requiring conditions			
Systemic autoimmune diseases, *n* (%)	1124 (3.1)	150 (11)	<0.001
Chronic pulmonary diseases, *n* (%)	3030 (8.5)	141 (10.3)	
Others, *n* (%)	31,613 (88.4)	1075 (78.7)	
Cumulative GC dose *, mg, mean ± SD	204.4 ± 150	190.1 ± 182.9	0.005
Comorbidities			
Hyperparathyroidism, *n* (%)	8 (0)	0 (0)	0.58
Hyperthyroidism, *n* (%)	538 (1.5)	41 (3)	<0.001
Hypothyroidism, *n* (%)	874 (2.4)	55 (4)	<0.001
Malignancy, *n* (%)	1046 (2.9)	81 (5.9)	<0.001

* Cumulative GC dose in the first 90 days after the initiation of GC therapy. GIOP glucocorticoid-induced osteoporosis, SD standard deviation, GC glucocorticoids.

**Table 3 medicina-58-00324-t003:** Associated factors for high-quality glucocorticoid-induced osteoporosis preventive care.

	Crude OR (95% CI)	*p*	Adjusted OR (95% CI)	*p*
Age				
<40 years	1 (ref.)		1 (ref.)	
40~49 years	2.57 (2.06–3.22)	<0.001	2.53 (2.02–3.17)	<0.001
50~59 years	3.86 (3.14–4.75)	<0.001	3.99 (3.23–4.92)	<0.001
60~69 years	4.5 (3.63–5.58)	<0.001	5.17 (4.16–6.43)	<0.001
≥70 years	7.14 (5.78–8.82)	<0.001	8.07 (6.5–10.03)	<0.001
Male	0.31 (0.27–0.35)	<0.001	0.26 (0.23–0.3)	<0.001
Residence				
Urban	1 (ref.)		1 (ref.)	
Rural	1.44 (1.22–1.71)	<0.001	1.19 (1–1.42)	0.046
Institution				
Tertiary/general hospital	1 (ref.)		1 (ref.)	
Primary care clinic/nursing hospital	0.53 (0.23–0.45)	<0.001	0.66 (0.57–0.75)	<0.001
Initial GC prescriber specialty				
Rheumatologist	1 (ref.)		1 (ref.)	
Non-rheumatologist internist	0.38 (0.27–0.53)	<0.001	1.07 (0.73–1.59)	0.722
Non-internist	0.32 (0.23–0.45)	<0.001	1.04 (0.7–1.53)	0.854
GC-requiring conditions				
Others	1 (ref.)		1 (ref.)	
Systemic autoimmune diseases	3.93 (3.28–4.7)	<0.001	3.08 (2.49–3.8)	<0.001
Chronic pulmonary diseases	1.37 (1.14–1.64)	0.001	1.04 (0.86–1.26)	0.708
Cumulative GC dose *, g	0.99 (0.99–1)	0.001	1.25 (0.91–1.71)	0.075
Comorbidities				
Hyperparathyroidism	N/A	0.957	-	-
Hyperthyroidism	2.03 (1.47–2.8)	<0.001	1.58 (1.13–2.21)	0.007
Hypothyroidism	1.68 (1.27–2.22)	<0.001	1.06 (0.8–1.42)	0.674
Malignancy	2.09 (1.66–2.64)	<0.001	1.59 (1.24–2.03)	<0.001

* Cumulative GC dose in the first 90 days after the initiation of GC therapy. OR odds ratio, GC glucocorticoids.

**Table 4 medicina-58-00324-t004:** Comparisons of epidemiologic data regarding glucocorticoid-induced osteoporosis preventive care in other studies.

	Country	Definition of Long-Term GC Use	BMD Test	Calcium/Vitamin D	Osteoporosis Medications	Associated Factor for GIOP Prevention
Albaum et al. [3] (*n* = 168,074)	Canada	Greater than or equal to two oral GC prescriptions dispensed and ≥450mg prednisolone equivalent over 6 months period	7% for men, 13% for women		12% for men, 30% for women	
Aagaard et al. [10] (*n* = 215)	USA	Prednisone (or its equivalent) at a daily dose of at least 5 mg for at least 1 month		Calcium 42%, vitamin D 37%	4%	Female ↑, aging ↑, rheumatologist ↑, more comorbid illness ↑, multiple medications ↑
Ettinger et al. [11] (*n* = 8807)	USA	Prescriptions for ≥2 g of prednisone (or equivalent) during any 12-month period			8.8%	Female ↑, aging ↑, higher GC exposure ↑, rheumatologist ↑, previous osteoporotic fracture ↑
Feldstein et al. [12] (*n* = 3031)	USA	Equivalent of >5 mg of prednisone per day for at least 90 days	9.8%		38%	
Curtis et al. [13] (*n* = 6281)	USA	Outpatient oral GC treatment on at least 60 days	33%	69%	37%	Female ↑, rheumatologist ↑, gastroenterologist ↓
Cruse et al. [14] (*n* = 370)	USA	Long-term oral prednisone use	44%	Calcium 51%, vitamin D 44%	24%	
Saag et al. [16] (*n* = 3125)	USA	≥7.5 mg/day of prednisone equivalent for >6 months	10~19%		50%	Female ↑, aging ↑, rheumatologist ↑
Majumdar et al. [17] (*n* = 17,736)	Canada	≥ 90 days of GC use	6%		22%	Female ↑, aging ↑, rheumatologist ↑
Trijau et al. [18] (*n* = 32,812)	France	≥7.5 mg of prednisone equivalent per day during at least 90 days	8%	18%	12%	Female ↑, aging ↑, rheumatologist ↑, gastroenterologist ↑, internist ↑, higher mean GCc dose ↑, RA ↑, autoimmune disease ↑, IBD ↑
Soen et al. [19] (*n* = 25,569)	Japan	GIOP risk score ≥ 3			51.8%	

GC glucocorticoids, GIOP glucocorticoids-induced osteoporosis, RA rheumatoid arthritis, IBD inflammatory bowel disease.

## Data Availability

Data from this study are available on request to the corresponding author of the study.

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
