# Peer review of "Status of Glucocorticoid-Induced Osteoporosis Preventive Care in Korea: A Retrospective Cohort Study on the Korean National Health Insurance Service Database"

_medicina, 2022, doi:10.3390/medicina58020324_

Round 1

Reviewer 1 Report

Byung Wook Song et al. investigated the status of glucocorticoid-induced osteoporosis preventive care in Korea: a nationwide population-based retrospective cohort study using the Korean National Health Insurance Service database. The study is interesting, but several study concerns should be mentioned.

  1. Osteoporosis is a disease that is characterized by low bone mass with microarchitectural disruption and skeletal fragility, resulting in an increased risk of fracture. In clinical practice, it is particularly relevant for the aging population, where fragility fractures, reduced glomerular filtration rate (GFR), and low bone mineral density (BMD) are more prevalent. There are some reasons should be presented why this differentiation is important, not the least of which is that management of osteoporosis differs vastly from treatment of other bone diseases in patients with chronic kidney disease (CKD).
  2. How to select the adjusted factors in a multivariable model? Please add CKD as one of the covariates in this study?
  3. The author should illustrate the patients' characteristics differences between the use of high and low dose of steroid. However, a correlation matrix between clinical factors and osteoporosis would also be interesting to find the possible clinical effect on glucocorticoid-induced osteoporosis.

Author Response

1. Osteoporosis is a disease that is characterized by low bone mass with microarchitectural disruption and skeletal fragility, resulting in an increased risk of fracture. In clinical practice, it is particularly relevant for the aging population, where fragility fractures, reduced glomerular filtration rate (GFR), and low bone mineral density (BMD) are more prevalent. There are some reasons should be presented why this differentiation is important, not the least of which is that management of osteoporosis differs vastly from treatment of other bone diseases in patients with chronic kidney disease (CKD).

: We appreciate your kind comment. Unfortunately, as Korean National Health Insurance Service database do not provide information regarding GFR, it was impossible to analyze the difference in osteoporosis treatment according to the presence or absence of CKD or the stage of CKD. This notion is considered as one of the limitations of our study.

2. How to select the adjusted factors in a multivariable model? Please add CKD as one of the covariates in this study?

: We selected adjusted factors in a multivariable model based on similar previous studies (Population-based trends in osteoporosis management after new initiations of long-term systemic glucocorticoids (1998-2008). J Clin Endocrinol Metab 2012, 97, 1236-1242, Osteoporosis prevention among chronic glucocorticoid users: results from a public health insurance database. RMD Open 2016, 2, e000249) and clinical relevance. As mentioned above, Korean National Health Insurance Service database do not provide information regarding GFR and it was impossible to include CKD in a multivariable model in the present study.

3. The author should illustrate the patients' characteristics differences between the use of high and low dose of steroid. However, a correlation matrix between clinical factors and osteoporosis would also be interesting to find the possible clinical effect on glucocorticoid-induced osteoporosis.

: In Korean GIOP guideline, very high dose of c is defined as ≥30 mg/day and a cumulative dose of >5 g in the past year. In this study, as cumulative glucocorticoids was low, thus it is difficult to compare clinical characteristics between low and high glucocorticoids dose. In our data, only age and cumulative glucocorticoids dose was continuous variables, it is not considered meaningful to analyze only the correlation between these two variables. In addition, Korean National Health Insurance Service database do not provide results of bone mineral density test such as dual energy x-ray absorptiometry, we could not conduct the correlation analysis between bone mineral density and cumulative glucocorticoids.

Reviewer 2 Report

Congratulations on the paper, it provides useful information for clinical practice. It is a topic of health and social impact 1) If you want you could change the title: "Status of glucocorticoid-induced osteoporosis preventive care in Korea:
a retrospective cohort study on the Korean National Health Insurance Service database"
2)
In the introduction to line 41 you could add the following reference: "Bone damage after chemotherapy for lymphoma:
a real word experience" Mancuso S. Scaturro D. Santoro M......
3)
In line 85: why did you choose age 20? Do you have any support references?
4)I
n line 88 there are no references on the posology. You should specify the dosage and whether they are equivalent doses
5)It is necessary to check the tables and references in the text. 
6)
In the flochart the inclusion criteria do not match the text. Check it out. 

7) In the discussion from line 248 it is not necessary to insert the percentages because they are already in the results. Change the form of the discussion

8)  why did you indicate table 4 in line 261?

Author Response

1) If you want you could change the title: "Status of glucocorticoid-induced osteoporosis preventive care in Korea: a retrospective cohort study on the Korean National Health Insurance Service database"
: We agree with your kind comment. We change the title of manuscript as you suggest.

2)In the introduction to line 41 you could add the following reference: "Bone damage after chemotherapy for lymphoma:a real word experience" Mancuso S. Scaturro D. Santoro M......
: We added the reference as you suggest.

3) In line 85: why did you choose age 20? Do you have any support references?
: Previous study by Majumdar et al. (Population-based trends in osteoporosis management after new initiations of long-term systemic glucocorticoids (1998-2008). J Clin Endocrinol Metab 2012, 97, 1236-1242) included adults 20 year of age and older, which support our inclusion criteria regarding age. 

4)In line 88 there are no references on the posology. You should specify the dosage and whether they are equivalent doses
: In line 88, we included any dose of oral GCs. Thus, we added the word “any dose” in this sentence. Cumulative glucocorticoid dose in our study refers to the dose equivalent to prednisolone (we added this notion in line 130).

5) It is necessary to check the tables and references in the text. 
: Thank you for your comment. Our review did not find anything particularly wrong with the manuscript. We would appreciate it if you could let me know where the error is.

6) In the flochart the inclusion criteria do not match the text. Check it out. 

: We corrected the figure of flow chart. We investigated subjects receiving oral GCs during 2011~2012.

7) In the discussion from line 248 it is not necessary to insert the percentages because they are already in the results. Change the form of the discussion

: We changed this sentence as you suggest. 

8)  why did you indicate table 4 in line 261?

: We deleted “table 4” in line 261.

Round 2

Reviewer 1 Report

All concerns have been clarified.

Reviewer 2 Report

The changes were made correctly. Good job!